# Proteomics Analysis of the Polyomavirus DNA Replication Initiation Complex Reveals Novel Functional Phosphorylated Residues and Associated Proteins

**DOI:** 10.3390/ijms25084540

**Published:** 2024-04-21

**Authors:** Rama Dey-Rao, Shichen Shen, Jun Qu, Thomas Melendy

**Affiliations:** 1Department of Microbiology & Immunology, Jacobs School of Medicine & Biomedical Sciences, University at Buffalo, State University of New York at Buffalo, Buffalo, NY 14203, USA; 2Department of Pharmaceutical Sciences, University at Buffalo, State University of New York at Buffalo, Buffalo, NY 14214, USA

**Keywords:** DNA replication, SV40 LT, Polprim, RPA, phosphorylation, proteomics, bioinformatics, DNA virus, network analysis, transcription factor

## Abstract

Polyomavirus (PyV) Large T-antigen (LT) is the major viral regulatory protein that targets numerous cellular pathways for cellular transformation and viral replication. LT directly recruits the cellular replication factors involved in initiation of viral DNA replication through mutual interactions between LT, DNA polymerase alpha-primase (Polprim), and single-stranded DNA binding complex, (RPA). Activities and interactions of these complexes are known to be modulated by post-translational modifications; however, high-sensitivity proteomic analyses of the PTMs and proteins associated have been lacking. High-resolution liquid chromatography tandem mass spectrometry (LC–MS/MS) of the immunoprecipitated factors (IPMS) identified 479 novel phosphorylated amino acid residues (PAARs) on the three factors; the function of one has been validated. IPMS revealed 374, 453, and 183 novel proteins associated with the three, respectively. A significant transcription-related process network identified by Gene Ontology (GO) enrichment analysis was unique to LT. Although unidentified by IPMS, the ETS protooncogene 1, transcription factor (ETS1) was significantly overconnected to our dataset indicating its involvement in PyV processes. This result was validated by demonstrating that ETS1 coimmunoprecipitates with LT. Identification of a novel PAAR that regulates PyV replication and LT’s association with the protooncogenic Ets1 transcription factor demonstrates the value of these results for studies in PyV biology.

## 1. Introduction

The polyomavirus (PyV) uses a major viral regulatory protein, large T-antigen (LT), that promotes viral proliferation [1]. LT is also involved in regulation of timing of the infection cycle, repressing transcription of viral early genes, stimulating expression of viral capsid proteins, and altering transcription of many cellular genes, thus preparing the host cells for substantial replication of the infecting virus [2,3,4]. It achieves that by binding to crucial host cell cycle regulators and tumor suppressors [5,6] as well as activating the transcription of viral late mRNA [7,8] and hinders histone deacetylation, leading to an overall activation of transcription [9]. LT is the only viral protein required for PyV genome replication, being the origin-binding protein and helicase, as well as recruiting the cellular factors required for replication of the viral DNA. All these interactions make LT an interesting target/tool in the search for compounds with antiviral and/or antiproliferative activities designed for the management of polyomavirus-associated diseases [10,11].

A schematic depiction of the elongating PyV DNA replication fork (Figure 1) shows LT, which, after recognizing the SV40 origin, acts as the hexameric AAA+ DNA helicase and recruits the DNA polymerase alpha-primase (Polprim) (a heterotetramer of 180, 68, 58, and 48 kDa) complex and the single-strand DNA binding Replication protein A (RPA) (a heterotrimer of 70, 32, and 14 kDa) complex, to drive replication of SV40 genomes [2,3,4,11,12,13,14,15,16,17,18,19,20,21,22].

All other factors required for initiation, elongation, and termination of SV40 DNA replication are provided by the host. It is well known that mutual interactions between LT, Polprim-, and RPA-complexes are critical for PyV DNA replication to be initiated: in origin binding, unwinding the origin of replication, helicase interaction with RPA, and loading RPA onto ssDNA, as well as for RNA-DNA primer synthesis [18,23,24,25,26,27,28,29,30,31,32,33]. The clamp loader/sliding clamp (RFC/PCNA) then recruits DNA polymerase delta to carry out processive synthesis of both strands [22,34,35]. Interactions of LT, Polprim, and RPA are known to be modulated by highly regulated and evolutionarily conserved PTMs that play a pivotal role in DNA replication, DNA repair, paused replication fork stabilization, or other yet unelucidated functions [36,37,38,39]. Although SV40 is the best-known eukaryotic DNA replication system [40,41,42], more remains to be understood about the subtleties and regulation of this process, including the inhibition of SV40 DNA replication in *trans* that occurs after treatment of cells with DNA-damaging agents [43,44,45,46,47]. Elucidating details of PyV DNA replication and its regulation may provide novel avenues to inhibit PyV DNA replication, potentially leading to PyV antivirals for use against progressive multifocal leukoencephalopathy and kidney nephropathy in immunocompromised hosts [10,11,48].

The paucity of highly sensitive proteomic studies on LT, Polprim-, and RPA-complexes has led us to apply immunoprecipitation (IP) combined with mass spectrometry (LC–MS/MS) technology (IPMS) [49] to identify phosphorylated amino acid residues (PAARs) and associated proteins for each of them. We identified 602 PAARs for the 8 polypeptides of LT, Polprim, and RPA, of which 479 had not previously been identified: 82 PAARs on SV40 LT, 305 on the Polprim heterotetrametric complex, and 92 on the RPA heterotrimeric complex. Using phosphomimetic mutation of one of the novel phosphorylated aa residues detected on LT in this study, we have recently demonstrated a dramatic decrease in DNA replication functions of SV40 Large T-antigen both in vitro and in cell culture [50]. The expansion of our understanding of PAAR sites provides new avenues to investigate regulation of important viral–host interactions with implications for devising targeted therapeutic strategies.

An in silico bioinformatic-based approach was used to analyze the co-immunoprecipitated (coIP-ed) proteins and revealed an enriched transcription-related GO process network within the SV40 LT associated proteins, notably distinguishing it from Polprim and RPA, and consistent with the known role of LT as a major transcriptional regulator. The curated 711 nonredundant novel protein associations with either one/two/all three complex clusters formed a “DNA Replication Signature” that included common molecules, networks, and biological processes linked to DNA replication and cell cycle_S phase, validating our proteomic dataset. The transcription factor (TF) TP53 was the most overconnected reaction hub, mirroring its regulatory significance to our replication-related dataset. Although the TF ETS proto-oncogene 1 (ETS1) was absent (‘hidden”) from our IPMS dataset, it was the most significantly overconnected TF to our dataset by in silico network analysis, indicating a potential role in the regulation of viral metabolism. While the mechanism of LT-dependent transcriptional activation remains ambiguous, the discovery of the overconnected ETS1 (an important activating cellular transcription factor) led us to question whether PyV LT might target ETS1. Both Ets1(~55 kDa) and p53 (~53 kDa) were confirmed to co-IP with LT from HEK293T cell lysates; the associations were demonstrated to be independent of DNA. Corroborating the p53/TP53 association with LT, both through IPMS and coIP, establishes the validity and robustness of our experimental and analytical methods due to the previously described roles of this critical interaction of p53 in viral replication and cellular transformation. Future studies will investigate whether the direct association of Ets1 with LT also plays an important role in viral transcription and/or cellular transformation. Our data provide substantial additional novel information on both PAARs and proteins associated with SV40 LT and the cellular Polprim- and RPA-complexes. Overall, we corroborated the previously described interactions of p53/TP53, POLA1, POLA2, RPA, and the association of TBX2 with LT by coIP and immunoblotting, as well as IPMS. We discovered Ets1 to be a TF overconnected to our dataset and demonstrated that it, too, is directly associated with LT by coIP, implicating the functional significance of Ets1 in PyV biology. Furthermore, we characterized the specific function of one of the novel PAARs (T518) in DNA replication (both in vitro and in cultured cell) [50]. Results from this study, thus, provide many avenues for future research in DNA replication, transformation, transcription, and other viral and host processes.

## 2. Results and Discussion

### 2.1. PAARs on SV40

To initiate viral DNA replication, LT requires specific contacts with the host Polprim and RPA complexes to carry out primer synthesis at the viral replication fork [24,25,51,52,53] that is regulated by PTMs. As a measure of phosphorylation both in Control (untreated/Cont) or etoposide (ETO)-treated cells by the DDR pathways, we used the overall SQ/TQ phosphorylation (a preferred ATM/ATR/DNA-PK target sequence) as a proxy. Immunoblotting for phospho-SQ/TQ of HEK293T cell extracts showed a slight increase in overall SQ/TQ phosphorylation in ETO-treated versus Cont in both whole (extr) or LT IP-depleted (depl) extracts, as well as in the eluted proteins from IP reactions (IP-LT, Figure 2A: compare lanes 1, 2, and 5 versus 3, 4, and 6).

Silver-staining of LT-IP proteins (lanes 7, 8) demonstrated a ~90 kDa band that coincided with SV40 LT by immunoblotting with specific antibodies (lanes 9–11). We compared the LT band in ETO versus Cont IP lanes (lanes 5, 6) and observed that while the protein amounts remained similar (by silver stain, lanes 7, 8) SQ/TQ LT phosphorylation was ~1.7-fold higher (ImageJ version 1.54g) in the ETO-treated as compared to Control (compare lane 5 to 6), indicating increased phosphorylation of LT by Phosphatidylinositol 3-kinase-related kinases (PIKKs) due to DNA damage.

Protein phosphorylation is used by cells to transiently change activity of enzymes, interactions, localizations, conformations, or even to target them for destruction [38,54,55]. Here, we used high-resolution MS2 to catalog the phosphorylation sites on the three major protein factors in the PyV DNA replication fork. The large number of PAARs identified in our study is consistent with the abundance of phospho-sites identified on many proteins using advances in mass spectrometric techniques in recent years [38,56].

A representative sequence map of SV40 LT shows the tryptic peptides (detected by MS) with 82 (Cont) and 85 (ETO) PAARs identified (Figure 2B) out of the total 104 Serine (S), Threonine (T), and Tyrosine (Y) that could potentially be phosphorylated (Figure 2B). There is a >90% coincidence of PAARs on LT from both treatments, with phosphorylated S378, T379, and S381 found only for ETO but not Cont A; a total of 59 PAARs were predicted in silico using bioinformatics tools in Netphos v3.1 (Appendix A, sheets 1–2). Figure 2C shows the graphical output. Our MS2 analysis identified several more phosphorylated S, T, and Ys than the predicted 59 by Netphos; also, there were 8 potential PAARs included on one undetected tryptic peptide in our MS analyses. Although we found a 1.7-fold higher phosphorylated LT band in ETO compared to Cont by immunoblotting, the MS method used, although highly sensitive, cannot quantitate differences between the two treatments. We surmise that since this MS analysis is not quantitative, that even low levels of DDR activation (such as that seen in unperturbed S phase cells) are sufficient to produce detectable phosphorylation on DDR phosphorylation sites in untreated cells. One of the novel phosphorylated aa residues, T518, detected on LT by this study, was investigated, and phosphomimetic mutation of this site was found to mediate dramatic inhibition of viral DNA replication [50], validating that at least one of our newly identified PAAR sites on LT mediates an important biological function.

The previously reported PAARs on LT were found to cluster into two small regions, one group closer to the amino terminal and the other to the carboxy terminal end of the protein [36,57,58,59]. It has previously been shown that activation of the LT double hexamer on SV40 origin DNA requires a unique phosphorylation state of LT: modified T124, and unmodified S120 and S123 [58,60,61,62,63,64,65]. However, these three residues were not identified in this report as they occur in a segment of the protein that was not detected by MS analyses. Of the previously reported 13 phosphorylated aa residues on LT, we verified 3 with high confidence (HC) and 4 with low confidence (LC), thus corroborating 54% of the previously published data (70% of those for which we obtained MS data). We add 82 novel PAARs on SV40 LT determined with HC (Appendix A, sheets 1–3).

#### Proteins coIP-Ed with SV40 LT

We identified 366 and 278 proteins associated with SV40 LT from Cont and ETO-treated extracts, respectively (Appendix A, sheets 1–3). The Venn diagram (Figure 2(D-1)) shows a substantial overlap (247 proteins) common to both Cont and ETO, with 119 and 31 unique to each, respectively, totaling 397 nonredundant proteins that were associated with LT. Our work confirmed 23 of the 126 previously reported LT-associated proteins, while discovering 374 additional proteins (Figure 2(D-2)). We have added a substantial number of novel LT-associated proteins compared to the previous state of knowledge (Appendix A, sheets 1–5). Further characterization of the relevance and functional/regulatory significance of the LT-associated proteomics data was accomplished following a bioinformatics-based approach and is discussed below.

### 2.2. PAARs on Polprim-Complex

Cell extracts were prepared, subjected to IP for the large catalytic subunit of Polprim, and the IP-fractions were resolved using SDS-PAGE and stained with silver or immunoblotted for phospho-SQ/TQ, as described above for SV40 LT. As before, an overall difference between levels of phosphorylation at DDR SQ/TQ sites was detected in DNA-damaged whole cell- or depleted-extracts (ETO versus Cont). Silver-staining of the IP eluted proteins (Figure 3A, lanes 7, 8) revealed almost no differences between Cont and ETO of the ~180 kDa band identified to be the catalytic subunit POLA1 by immunoblotting with protein-specific antibody (lanes 9, 10).

The ratio of this phosphorylated (~180 kDa) band (compare lane 6 to 5) was 1.1 (ImageJ), suggesting that this catalytic subunit of DNA polymerase was not highly phosphorylated in response to DNA damage. IPMS led to identification of all four Polprim subunits in the IP-fractions, and like the SV40 LT sequence map described above, only one representative with modifications is provided for each: POLA1(p180, catalytic subunit) and POLA2 (p68, polymerase auxiliary subunit), PRIM1 (p48 primase subunit) and PRIM2 (p58 primase subunit) (Appendix A). We found a very high coincidence of PAARs between Cont and ETO for each subunit, with a few exceptions (details in Appendix A). We detected a total of 184, 79, 62, and 75 phospho-S, T, and Ys on POLA1, POLA2, PRIM1, and PRIM2 subunits, respectively, by IPMS (Appendix A, sheets 1–4) and predicted 124, 81, 35, and 48 phospho-S, T, and Ys in silico by Netphos v 3.1 (Appendix A, sheet 5) with graphical output (Appendix A). For most subunits of the Polprim-complex, we detected more PAARs than those predicted; however, ~10–15% in silico predicted sites were not identified in our MS, primarily due to the residues being on un- or poorly detected tryptic peptides.

Comparing previously published data gleaned from PhosphoSitePlus [66], our analysis was able to confirm all previously reported phosphorylated residues for POLA1 with HC (100%), 22 out of 24 with HC and 2 (Y526, Y528) with LC on POLA2 (100%), 13 out of the 15 with HC (87%) and 2 (S123, S124) with LC on PRIM1 (100% in total), and, finally, 19 out of 20 with HC and 1 (S492) with LC for PRIM2 (100%), thus corroborating all previously published PAARs for all subunits of the Polprim-complex (Appendix A, sheets 6–9). The high level of coincidence and overlap of our experimental results with previous reports further validate our data. Overall, we were able to add 155, 48, 46, and 56 novel phosphorylated S, T, and Ys with HC on the respective four subunits of Polprim.

It is well known that Polprim is essential for PyV DNA replication with at least three subunits that directly interact with LT [21]. Cell-cycle-dependent phosphorylation of DNA polymerase alpha inhibits DNA synthesis and may affect Polprim’s physical interactions with either other replicative proteins or with DNA [67]. Polprim is most active when its regulatory subunit (POLA2) is phosphorylated by CyclinA-Cdk2, and the N-terminal end of its catalytic subunit (POLA1) is not [39,68,69]. Of the known PAARs on DNA polymerase alpha, the site at the N-terminal end of POLA1, T174, and the potential Cdk recognition sites on POLA2, S141, S147, S152, and T156, as well as three putative phospho-sites T115, T127, and T130, were confirmed by our work. Identification of the additional PAARs on the Polprim-complex will likely prove important in further understanding of its roles in DNA replication.

#### Proteins coIP-Ed with Polprim-Complex

The Venn diagram of coIP-ed proteins with Polprim-complex (Figure 3(B-1)) shows 144 overlapping between Cont- and ETO-treatments with 308 and 20 unique to each, respectively (Appendix A, sheets 1–3). Our study confirmed 19 out 267 previously reported Polprim-associated factors, while identifying 453 additional ones, thus adding a sizeable number of novel Polprim-associated proteins compared to what was previously known (Figure 3(B-2) and details in Appendix A, sheets 4–5).

### 2.3. PAARs on RPA-Complex

As shown above for SV40 LT, there is an increase in overall SQTQ phosphorylation on ETO treatment (DNA-damaged) over Control (Figure 4A lanes 1–4).

The co-precipitated proteins eluted from beads for both treatments were stained with silver (lanes 7, 8). The silver-stained bands at 70 and 32 kDa were identified as RPA1 and RPA2, respectively, by immunoblotting. The slower migrating phosphorylated RPA2 (p-RPA2) is seen in ETO-treated IP-RPA2 eluate by SQ/TQ antibodies (lane 6), silver stain (lane 8), and RPA-specific antibodies (lanes 13), as well as in ETO-treated whole/depleted cell extracts (lanes 11–12) but not in Cont IP fraction (lanes 5 and 7), or Cont whole/depleted cell extracts (lanes 9–10). While RPA1 and the nonphosphorylated RPA2 band show equivalent levels of phosphorylation in Cont and ETO treatments (ratios close to 1.0 by ImageJ) (compare lanes 5 and 6), there was a 23-fold increase for the slower-migrating phospho-RPA2 (p-RPA2) band quantitated from ETO versus Cont (compare lane 6 to 5), supporting the known role of RPA2 as being a major phospho-target of the ATM/ATR kinases [43,70]. ATR-dependent phosphorylation is required to inhibit cellular DNA replication in response to DNA damage, although the mechanism remains unknown [71].

Our IPMS method identified all three of the RPA subunits in the IP-eluates using RPA-specific antibodies, and a sequence map was generated for each subunit: RPA1 (70 kDa), RPA2 (32 kDa), and RPA3 (14 kDa) (Appendix A). Sequence coverage for detected tryptic peptides by LC–MS/MS was 100% for all three RPA subunits. We scored the PAARs for each subunit and demonstrated the few differences between two treatments in Appendix A, sheets 1–3. It is notable that while we can easily detect the phosphorylated state of the RPA2 (p-RPA2) band between Control and ETO-treated IP by both immunoblotting and silver staining, our MS analysis was unable to detect any appreciable differences between them quantitatively. As discussed earlier, this is probably due to the highly sensitive, but nonquantitative, LC–MS/MS detection method used. Thus, even infrequent phosphorylation events, such as DDR-mediated phosphorylation events, in Control cells could be detected. The PAARs identified by IPMS are presented in context of the in silico predictions of 70, 34, and 9 phospho-S, T, and Ys on the 3 subunits, respectively, via Netphos v3.1 (Appendix A, sheet 4) with the graphical output included in Appendix A. We discovered a total of 81, 31, and 13 PAARs by IPMS on RPA1, RPA2, and RPA3, respectively.

Upon comparing previously published phosphorylation data [66] for the three subunits of the RPA-complex, our analysis was able to confirm 23 out of 24 phosphorylated residues with HC and 1 (S135) with LC (100%) for RPA1, 6 out of 17 with HC and 2 (S12 and S33) with LC for RPA2 (47%), and 4 out of 4 previously reported phosphorylated amino acid residues for RPA3 (100%) (Appendix A, sheets 5–7). We thus corroborate all previously published phospho-data for both RPA1 and RPA3 and up to 47% for RPA2. We were able to corroborate the well-known S33 to be phosphorylated on RPA2, albeit with low confidence [72,73]. We therefore add 58, 25, and 9 novel phosphorylated S, T, and Ys with HC on the respective three subunits of the RPA-complex.

The functional relevance of PAARs identified by IPMS analysis is validated by (1) high percentage of previous PAARs corroborated on all 8 polypeptides of SV40 LT, Polprim, and RPA; (2) the overall coincidence between our experimental findings and in silico predictions; and (3) we found one PAAR on LT (T518) to be directly involved in viral DNA replication. Therefore, the potential importance and/or relevance of the other novel PAARs identified in this study holds promise and will be investigated in future studies.

#### Proteins coIP-Ed with RPA-Complex

LC–MS/MS analysis of the immunoprecipitated fractions using anti-RPA antibodies identified a total of 225 coIP-ed proteins in both Cont and ETO. The Venn diagram (Figure 4(B-1)) shows that 43 out of the 45 RPA associated proteins in ETO overlap with Cont. There were 180 unique coIP-ed proteins to Cont and 2 to ETO (Appendix A, sheets 1–3). Our study was able to confirm 42 of the previously reported 673 RPA-associated factors, while identifying 183 novel protein associations (Figure 4(B-2) and Appendix A, sheets 4–5).

The low level of similarity of protein associations that we found for all three LT, Polprim, and RPA compared to previous studies might be due to the variability in methodologies as well as the transient nature of protein associations/interactions during DNA replication. To examine the functional relevance and regulatory significance of the large proteomic dataset, we further investigated it via several bioinformatics-based tools.

### 2.4. GO Enrichment Analysis

GO enrichment analyses were based on the function of all coIP-ed proteins with the three SV40 LT-, Polprim-, and RPA-complexes separately. A significant enrichment in a transcription-related process network was observed with SV40 LT-associated proteins (Figure 5, red box).

The highly enriched “transcription-mRNA processing” network included 50 network objects (proteins) that were uniquely and significantly associated with SV40-LT (*p*-value = 2.59 × 10^−21^, FDR 1.16 × 10^−19^), as opposed to Polprim and RPA (Appendix A). See Appendix A sheet 1–3 for statistics. This is consistent with the functionality of the viral protein, LT, as a transcriptional regulator that is capable of recruiting and interacting with the host cellular transcription machinery to regulate several cellular processes, including SV40 DNA replication.

### 2.5. Creation and Characterization of the “DNA Replication Signature”

Removing redundancies and curating coIP-ed proteins with LT, Polprim, and RPA allowed a global investigation into the relevance of our data to SV40 DNA replication. We delineated this curated 711 nonredundant coIP-ed proteins/gene IDs as the “DNA Replication Signature” (Figure 6A).

After the initial set of in silico bioinformatics-based analyses to validate the curated signature related to DNA replication and cell cycle, subsequent network and interactome analyses were undertaken to identify potentially functional molecular elements and transcriptional regulators within and “hidden from the “DNA Replication Signature”. Details of proteins included within the signature (indicated by HGNC approved Gene IDs) along with highest spectral counts (a semiquantitative measure for protein abundance) in each IP reaction are available in Appendix A, sheets 1–2. The signature showed a considerable overlap (~42%) with previously reported host protein factors (along with viral LT) known to be associated with the SV40 DNA replication fork (Appendix A, sheet 3), validating our results. The well-studied “Cell cycle_S phase” was the most significant of nine networks associated with our dataset [68,75,76,77] (Appendix A). The significant biological processes enriched within this network were related to DNA metabolic process, DNA replication, cell cycle, and DNA damage and repair, among others (Appendix A), and included 26 proteins from our dataset. For statistics underlying the “Cell cycle_ S phase” network, see Appendix A, sheets 1–9.

The Venn diagram (Figure 6B) shows that 119 of the Gene IDs were shared between all three factors, with 86, 49, and 10 shared between LT-Polprim, Polprim-RPA, and RPA-LT, respectively, and 182 were unique to LT, 218 to Polprim, and 47 to RPA. A GO enrichment analysis (Figure 6C) of this curated dataset by protein function revealed three highly relevant protein classes within the dataset out of several other functional categories that failed to pass the cut-off values for significance. The three classes (enzymes, kinases, and transcription factors) are associated with low *p*-values (6.391 × 10^−34^, 0.01381, and 0.01014, respectively) and high z-scores (14.65, 2.506, and 2.56, respectively) and are prioritized since they contain the 204 most overconnected (and therefore essential) genes [78]. For statistics, see Appendix A, sheets 4, 5.

#### 2.5.1. Interactome Analysis

Interactome and network analyses of the IPMS curated dataset in the context of the larger human proteome in the metabase allowed for a deeper interpretation of the complex proteomic dataset. The three major replication fork complexes are known to be connected to one another directly for effective replication initiation [18,23,24,25,26,27,28,29,30,31,32,33] and were also found to coIP with each other. POLA1, POLA2, and RPA2 coIP-ed with LT, and the four subunits of Polprim and two subunits of RPA coIP-ed with Polprim (Appendix A, sheet 2 and Appendix A, sheet 1). All three factors with their subunits, used as seed nodes, were also observed to be connected to one another in an in silico interactome analysis (Figure 7).

The lower number of overall connections (30 interactions) associated with LT, as opposed to the cellular complexes RPA (196 interactions) and Polprim (211 interactions), to genes from within or outside our experimental dataset is to be expected since LT is a viral protein that can interact with factors within the cell and bring about viral replication but would not be expected to be as interconnected as cellular proteins. In this network, all three factors were connected by two proteins implicated in replication and cell cycle progression: (1) Histone Deacetylase 3 (HDAC3), known to catalyze the deacetylation of lysine residues of the core histones playing an important role in cell cycle progression [79], and (2) Specificity protein1 (SP1), a transcription factor, known to regulate expression of a large number of genes involved in cell growth [80]. The only connector protein observed between LT and Polprim in this network is Sirtuin1 (SIRT1) that was shown to bind with HDAC1 and HDAC3 to deacetylate and activate LT [79] acting as a gatekeeper of replication initiation [81]. Additionally, LT is observed to be connected to the RPA-complex (directly or indirectly as both inhibitor and activator) via three genes (TP53, EP300, and ATM) that have been implicated in replication and cell cycle progression. ATM serine/threonine kinase (ATM) is a member of the PIKK family involved in the phosphorylation of RPA2 as part of the DDR response that is essential for optimal SV40 replication in primate cells [82,83]. Mapping studies have established that the N-terminal half as well as the ssDNA binding domain of RPA1 are interaction sites for POLA1 and essential for polymerase primer synthesis, while the interaction with LT interfered with the RPA1-POLA1 interaction. The competition between LT and POLA1 for RPA might be playing a crucial role in switching POLA1 function from priming to DNA synthesis [28].

#### 2.5.2. Transcriptional Regulation Network Analysis

The most significantly overconnected network to our dataset using the “transcriptional regulation network” algorithm was named by the associated TF, ETS1 (*p*-value = 0, z-score = 524.25) (Appendix A, sheet 2). The merged network (with externally added LT) was connected to ~118 genes from within our dataset (Figure 8) and associated with cell cycle and apoptotic process among others (Appendix A, sheet 11).

We discovered that while ETS1 itself was absent in our experimental dataset, TFs such as TP53 (with 55 interactions) and T-box transcription factor 2 (TBX2 with 54 interactions) were from within our experimental dataset (black circles). For details and statistics underlying the network, see Appendix A, sheets 1–11. The tumor suppressor protein p53 is known to bind to SV40 LT and regulates cell division by preventing cells from growing uncontrollably [84,85]. The overconnectivity of TP53 and TBX2 from within our dataset and ETS1 hidden from it demonstrates the likely importance of all the three transcription regulators.

#### 2.5.3. Corroborating Known Interactions with SV40 LT

Immunoprecipitation with antibodies to SV40 LT from HEK293T cell lysates were immunoblotted with antibodies to POLA1, POLA2, RPA, and SV40 LT (Appendix A). This corroborated their well-established direct interactions that were initially identified by IPMS and immunoblotting (Appendix A, sheet 2 and Appendix A, sheet 1) [18,23,24,25,26,27,28,29,30,31,32,33]. These interactions form the basis for the interactome analyses (Figure 7). The association of p53 with LT is shown by coIP in Figure 2A (lane 11) as well as in Figure 9A (lanes 3, 4). This association is independent of DNA (Appendix A) and validates our proteomics dataset and the bioinformatics-based analyses reported here.

We further validated the co-IP of the third overconnected TF associated with the transcriptional regulation network: TBX2 (~72 kDa) coIPs with SV40 LT, and the association is free of DNA (Appendix A). Future work will reveal the importance of all three TFs in viral biology.

#### 2.5.4. Identification of a Novel LT-Associated Factor: Ets1

Since ETS1 was the most significantly overconnected and largest hub in the TF network, but “hidden” (absent) from our dataset, we questioned whether ETS1 might also associate with LT. Many interacting proteins present at low levels or with low affinity can go undetected in LC–MS/MS datasets due to signals being below the level of detection. To investigate the association of ETS1 with SV40 LT, we performed coIP experiments using LT antibodies with lysates from both Control and ETO-treated HEK293T cells. Immunoblots were probed with LT, Ets1, and p53 antibodies (Figure 9A) and clearly demonstrated that Ets1 (~55 kDa) coIP-ed with LT, very close to TP53/p53 (~53 kDa) (lanes 3, 4).

To ascertain that the association of LT and Ets1 is independent of DNA, the HEK293T cell lysates were digested with Benzonase-nuclease treatment before the IP reaction (Figure 9B). Benzonase treatment was able to remove DNA very effectively (lanes 3, 5 vs. 2, 4). Ets1 and p53 both continued to coIP with LT at similar levels after Benzonase treatment (Figure 9C, lanes 3, 4, and Appendix A). Hence, this previously unreported association of Ets1 with LT is independent of DNA. ETS proto-oncogene 1, transcription factor (Ets1) is a member of the ETS family of evolutionarily conserved sequence-specific DNA-binding transcriptional activators that play a major role in tumor progression [86,87] and in diverse cellular processes such as proliferation, differentiation, lymphoid development, motility, invasion, and angiogenesis that are likely to be dependent on specific protein interactions [88,89]. Since SV40 LT is a major transformation-inducing protein and is known to target several tumor suppressors such as tumor suppressor protein p53 (TP53), RB transcriptional corepressor 1 (RB1), and mitotic checkpoint serine/threonine kinase (BUB1), among others, this finding will likely fuel further study of SV40 Large T antigen targeting the transcriptional protooncogene, ETS1, to investigate its specific regulatory function in viral transcription and cellular transformation.

## 3. Materials and Methods

### 3.1. Cell-Culture and Lysate for IP

Human embryonic kidney 293T (HEK293T) cells (American Type Culture Collection (ATCC), Manassas, VA, USA) were grown at 37 °C with 5% CO_2_ in Dulbecco’s Modified Eagle’s Medium (DMEM) (Gibco) supplemented with 10% fetal bovine serum and 1% penicillin–streptomycin (Gibco). To induce DNA damage, 70–90% confluent cells were treated with 50 μM etoposide solution (Millipore- Sigma, St Louis, MO, USA) ) in DMSO (ETO) or were left untreated with only DMSO (Control or Cont) for 2 h at 37 °C. Cell lysates were prepared using NP40 lysis buffer (ThermoFisher Scientific, Carlsbad, CA, USA) containing 50 mM Tris-Cl (pH 7.4), 250 mM NaCl, 1% NP40, and 5 mM EDTA with protease and phosphatase inhibitors (ThermoFisher Scientific, Carlsbad, CA, USA Millipore-Sigma, St Louis, MO, USA), on ice, subjected to centrifugation in a microfuge for 15 min at 4 °C at 13,000 rpm. Protein levels were quantitated using BCA assay, flash-frozen in liquid nitrogen, and stored at −80 °C.

### 3.2. Antibodies

Monoclonal antibodies 101 or 419 to large T-antigen (LT), 1644 or 1645 (anti-DNA Polymerase alpha-: POLA1), 9 (anti-70 kDa RPA subunit, RPA1), 20 (anti-32 kDa RPA subunit: RPA2), and p53 were generated in our laboratory and purified on protein A Sepharose (GE- Thermofisher Scientific, Carlsbad, CA, USA). Commercial rabbit polyclonal antibodies to POLA1 were purchased from Abcam (Waltham, MA, USA), rabbit anti-Actin (β-actin, ACTB) (Millipore-Sigma, St Louis, MO, USA and rabbit anti-TBX2 (Proteintech, Rosemont, IL, USA ). We used two Ets1 antibodies for our LT-IP experiments: (1) mouse mAb (NBP2-2216, Novus, Centennial, CO, USA); (2) rabbit mAb (MA5-32732, Thermofisher Scientific, Carlsbad, CA, USA). All primary antibodies were used at 1:1000–5000 dilution for immunoblotting (IB). Rabbit polyclonal antibody (pSQ/TQ) (Cell Signaling Technology, Danvers, MA, USA): IB 1:500 in 3% BSA in TBST (50 mM Tris-HCl, 150 mM NaCl, 0.1% Triton-X100).

### 3.3. IP for ImmunoBlot

For immunoprecipitation, we followed the manufacturer’s protocol with the following modifications: a 50 µL slurry of Dynabeads Protein G (Invitrogen, Carlsbad, CA, USA Cat # 10003D) was used to immunoprecipitate proteins from ~500 µL ETO and DMSO-treated HEK293T cell extracts (1.3 mg/mL protein concentration) bound to ~1.0 µg LT, Polprim, and RPA-specific monoclonal antibodies. After extensive washing, the proteins were eluted from the beads in NuPAGE LDS reducing sample buffer (Thermofisher Scientific, Carlsbad, CA, USA)., heated to 90 °C, and resolved on 4–12% or 4–20% Novex Tris-glycine gels for 1.5 h at 130 volts. Gels were transferred to a nitrocellulose membrane using iBlot2 gel transfer using template program P0. Membranes were blocked for 1 h at room temperature with 5% nonfat dry milk (NFDM) in TBST, and washed 3 × 5 min with TBST. Appropriate primary antibodies (either 1 h at RT or O/N at 4 °C with shaking) were used to probe the membranes, washed 3 × 5 min with TBST, and placed in HRP-linked anti-rabbit or anti-mouse secondary antibody (diluted to 0.5 μg/mL in TBST for 1 h at RT). Membranes were washed 3 × 5 min with TBST and developed using chemiluminescent substrate (Advansta, San Jose, CA, USA) and imaged using a ChemiDoc (Bio-Rad, Hercules, CA, USA). Fiji, the open-source platform for ImageJ v1.54g (NIH), was used to quantify bands.

Validation: To examine the role of nucleic acids on the co-immunoprecipitation of LT and ETS1, cell lysates from HEK293T cells were treated with Benzonase (800 U/mL) (Millipore-Sigma, St Louis, MO, USA) in the presence of 1 mM MgCl_2_ for 2 h at room temperature while shaking. Protein lysates were checked for nucleic acid degradation on agarose gel stained with ethidium bromide (EtBr) before adding LT-specific antibodies to the enzyme digested lysate for immunoprecipitation with Dynabeads Protein G; proteins were resolved on 4–12% Novex Tris-glycine gels and WB was performed as above.

### 3.4. IP for Mass Spectrometry

LT, Polprim, and RPA-specific mAbs (approximately 100 µg of each ligand) were separately bound to 5 mg of M-280 Tosyl-activated Dynabeads (Invitrogen, Carlsbad, CA, USA Cat # 14203) O/N at 37 °C on a rotating wheel, washed twice in 1X PBS, and prepared according to the manufacturer’s protocol. Binding reactions were set up with the equivalent of 1 mg beads + ligand in PBS mixed with 1.3 mg of total protein (whole extracts of Cont and ETO-treated cells separately) at 3.0 mg/mL final concentration for 4 h at 4 °C with tumbling. The beads were washed with PBS three times and eluted with 0.2% formic acid (FA) 4× at RT. The elution was flash-frozen in liquid N2 and dried in a speed-vac. Portions of immunoprecipitations were boiled in reducing SDS loading buffer at 95 °C for 5 min, and the bound proteins were resolved using 4–12% Novex Tris-glycine gels and silver-stained (MS compatible) (ThermoFisher Scientific, Carlsbad, CA, USA). Trypsin digestions of the 0.2% formic acid (FA) elution were processed for mass spectrometric (MS) analysis at the Proteomics and Bioanalysis Core (PBC) Facility, New York State. The IP followed by MS-based proteomics analysis [90] is termed as IPMS in the paper. The solvent-eluted sample, resulted in peptide identifications and good signal-to-noise ratios.

### 3.5. Mass Spectrometry (MS)

Liquid chromatography–tandem mass spectrometry (LC–MS/MS) was carried out on a trapping nano-flow LC-Orbitrap MS system consisting of a Dionex Ultimate 3000 nano LC system, a Dionex Ultimate 3000 gradient micro-LC system with an WPS-3000 autosampler, and an Orbitrap Fusion Lumos mass spectrometer (ThermoFisher Scientific, San Jose, CA, USA). For preparation of lyophilized LT samples, 50 μL 0.5% SDS was added to each sample, and samples were sonicated for 30 s and vortex-mixed for 10 min to reconstitute protein. Protein reduction and alkylation was performed sequentially by addition of 2 μL 200 μM DTT and 4 μL 500 μM iodoacetamide (IAM), each with 45 min incubation in a covered thermomixer under 37 °C with constant shaking. Protein was then precipitated by two-step addition of 60 and 300 μL chilled acetone, incubated at −20 °C for 3 h, and centrifuged for 30 min at 18,000× *g* under 4 °C to pellet precipitated protein. Pelleted protein was gently rinsed with 400 μL methanol, decanted, and wetted using 45 μL 50 mM Tris-FA. A volume of 5 μL trypsin dissolved in 50 mM Tris-FA (0.25 μg/μL) was added to each sample, and trypsinization was performed under 37 °C overnight (~16 h) with constant shaking in a covered thermomixer. Trypsinization was terminated by addition of 0.5 μL FA, then the derived peptide mixture was centrifuged at 18,000× *g* under 4 °C for 30 min, and supernatant was transferred to vials for analysis.

A single injection of derived peptides was analyzed for each sample. A large-i.d. trapping column (300 µm ID × 5 mm) was implemented prior to nano LC column (75-μm ID × 100 cm, and packed with 3 μm Pepmap C18) separation for high-capacity sample loading, matrix component removal, and selective peptide delivery. Mobile phase A and B were 0.1% FA in 2% acetonitrile and 0.1% FA in 88% acetonitrile. The 180 min LC gradient profile was 4–13% B for 15 min; 13% to 28% B for 110 min; 28% to 44% B for 5 min; 44% to 60% B for 5 min; 60% to 97% B for 1 min; and isocratic at 97% B for 17 min. MS was operated under data-dependent acquisition (DDA) mode, with a maximal duty cycle time of 3 s. MS1 spectra were acquired in the *m*/*z* range 400~1500 under 120 k resolution with dynamic exclusion settings (60 s ± 10 ppm). Precursor ions were filtered by quadrupole using a 1 Th wide window and fragmented by high-energy C-trap dissociation (HCD) at a normalized collision energy of 35%. MS2 spectra were acquired under 15 k resolution in either Orbitrap (OT) or Ion Trap (IT).

### 3.6. Data Analysis

LC–MS raw files were searched by Sequest HT (embedded in Proteome Discoverer v1.4.1.14, ThermoFisher Scientific) against SV40 LT sequence. The search parameters included (1) precursor ion mass tolerance: 20 ppm; (2) fragment ion mass tolerance: 0.02 Da (OT)/0.8 Da (IT); (3) maximal missed cleavages: 2; (4) fixed modification: cysteine carbamidomethylation; (5) dynamic modification: methionine oxidation, peptide N-terminal acetylation (spontaneously occurring protein modifications, labeled as O and A in the protein PTM map; serine/threonine/tyrosine (S/T/Y) phosphorylation (labeled as P in the protein PTM map); (6) maximal modifications per peptide: 4. The acetylation of free amine groups is on any aa, including Lys (K) and Arg (R) termini, and occur spontaneously during experimental steps and may not be biologically meaningful unless validated. To avoid overinterpretation of the data, we do not include acetylated residues in our findings. Abbreviations: P = phosphorylated residues, A = acetylated residues. LT, Polprim, and RPA sequence coverage and peptide lists were exported from Proteome Discoverer. Results were consistent and stable for tryptic peptides of eluted proteins in solution rather than in-gel digestions. The lists of interacting proteins are referred to by Gene IDs (HGNC approved) encoding them for convenience and downstream bioinformatics-based analyses.

### 3.7. Bioinformatics-Based Analyses

#### 3.7.1. Prediction of PAARs by In Silico Analysis (Netphos v3.1)

Prediction of phosphorylated S, T, Y sites: The Netphos v3.1 server (https://services.healthtech.dtu.dk/service.php?NetPhos-3.1 accessed on 5 August 2021) was used to predict Ser-, Thr-, and Tyr- (S, T, Y) phosphorylation sites for the full sequences of SV40 LT, PolPrim- (4 subunits) and RPA- (3 subunits) complex. The complete “native” sequence of each protein/subunit was submitted in FASTA format for best prediction, and all three residues (S, T, Y) were selected for the calculation of phosphorylation potential. The default score of ≥0.5 was used as the cutoff for a positive prediction. The server predicts S, T, or Y phosphorylation sites on eukaryotic proteins using ensembles of neural networks. Both generic and kinase specific predictions are performed. The kinase-specific predictions are based on consensus cleavage site for 17 kinases (ATM, CKI, CKII, CaM-II, DNAPK, EGFR, GSK3, INSR, PKA, PKB, PKC, PKG, RSK, SRC, cdc2, cdk5, and p38MAPK) that include two of the three PIKK family: Ataxia telangiectasia mutated serine /threonine kinase (ATM) and DNA-dependent protein kinase catalytic subunit or protein kinase, DNA activated, catalytic subunit (DNAPK or PRKDC) involved in DNA damage response (DDR) kinase phosphorylation of several proteins [91]. To search for the previous literature for binding proteins and PAARs on SV40 LT, Polprim-, and RPA-complexes, we used the MetaCore database (https://portal.genego.com, accessed on 14 October 2021), PhosphoSitePlus (https://www.phosphosite.org/homeAction v 6.7.1.1powered by Cell Signaling Technology Beverly, MA, USA, accessed on 24 Februray 2022), and PUBMED (https://pubmed.ncbi.nlm.nih.gov/ accessed on 1 January 2021).

#### 3.7.2. GO Enrichment Analysis

Our IPMS experiments led to identifying coIP-ed proteins listed by Gene IDs approved by HUGO Gene Nomenclature Committee (HGNC) with the three major proteins SV40 LT, Polprim-, and RPA-complex at the replication fork. We mostly refer to the proteins by their HGNC approved gene IDs in this report. Each protein list pair (Cont and ETO) (Gene IDs) was uploaded into the web-based integrative software suite from GeneGO: MetaCore+MetaDrug^®^ version 22.2 build 70,900 database (Clarivate) (https://portal.genego.com accessed on 14 October 2021) and mapped to network objects for Gene Ontology (GO) enrichment analyses [92,93] with the whole human proteome in the background. Hypergeometric tests [94,95] were used to calculate statistical significance (as *p*-values and z-scores), in which the null hypothesis is that no difference exists between the number of genes falling into a given category in the target experimental gene list and the genome. Enrichment analysis consists of matching gene IDs of possible targets for the “common”, “similar”, and “unique” sets with gene IDs in functional ontologies in MetaCore.

To evaluate relevance in biological processes, pathways, and process networks, each pair (Cont and ETO) of LT, Polprim, and RPA-complex associated proteins was used in a function-based “comparative enrichment analysis” (enrichment by protein function, network building algorithms). The fundamental assumption is that relative connectivity of a gene mirrors its functional significance for the enriched biological processes. Relative connectivity is calculated as the number of interactions between the experimental genes with the genes on the experimental list normalized to the number of interactions it has with all genes in the human proteome in the MetaCore database (Metabase). We focused on the top 10 over-represented process networks (with *p*-value ≤ 0.01 and false discovery rate ≤ 0.01), which included at least 4 genes.

#### 3.7.3. “DNA Replication Signature” for Global Analyses

To obtain a global understanding of the biological context within the vast amount of proteomic data generated, we removed all redundancies in the paired lists of proteins (Gene IDs) (Cont and ETO) and made a list of proteins that were associated with one, two, or all three factors (SV40 LT, Polprim-, and RPA-complexes). We label this curated nonredundant list of 711 proteins (Gene IDs) that coIP-ed with any of the three factors as the “DNA Replication Signature”. Each entry was associated with the highest spectral counts (SC) detected among all three coIP-ed lysates, which is generally accepted as a semiquantitative measure for protein abundance in the IP eluate (cell lysates). Gene IDs of these 711 proteins were then imported into MetaCore and mapped to ~806 network objects (NOs). This increased number of NOs to proteins in the list is due to inclusion of complexes along with individual entries. This is also referred to as the “experimental dataset” and is activated in silico for bioinformatics-based network analyses with the whole human proteome in the background. We invoked the GO enrichment analysis by protein function that scored and ranked the most relevant significant protein functions related to the curated experimental dataset since these functional categories have overconnected genes that are the most essential for function [78].

#### 3.7.4. Network Analysis

The in silico network-based analyses leverage the manually curated knowledge within the MetaCore database [93]. We used different algorithms to drill down into our experimental dataset of 711 gene IDs (representing coIP-ed proteins with LT, Polprim-, and RPA-complex). To study the networks in the database that are most related to our experimental dataset, we used the keyword “DNA Replication” to execute a search of the knowledge base in MetaCore. The most significant of the nine networks associated with the term “DNA replication” was “Cell Cycle_S Phase”. We externally added SV40 Large T-antigen (not in the human proteome) and allowed the single interaction algorithm (up- and downstream) to retrace the network integrating the viral protein into the prebuilt mammalian Cell Cycle_S Phase network from the database that incorporated our experimental dataset as well.

#### 3.7.5. Interactome Analysis

To investigate connections with the three prioritized replication fork factors (SV40 LT, Polprim (4 subunits), and RPA (3 subunits)) that were used for the IPMS experiments, an interactome analysis was conducted. Using all subunits of the three factors as seed nodes, we activated the “build networks” algorithm “expand by one interaction” (up- and downstream) to create merged biological networks. These networks are built on the fly and are unique for our activated experimental dataset (gene content as well as spectral counts) with the human proteome in the background. No low-trust interactions were included, and while retaining the mi-RNA connections we chose not to show names of interacting compounds and drugs. Investigations to address these novel findings are potential areas for future studies.

#### 3.7.6. Transcription Regulation Network Analysis

To explore significant transcription factors (TFs) that are overconnected to proteins in our dataset associated with biological processes such as DNA replication and cell cycle, we used the transcription regulation network algorithm. Networks were built with a variant of shortest paths algorithm with relative enrichment of our activated experimental dataset combined with a relative saturation of networks related to GO processes such as cell cycle, DNA replication, and cell death. The networks were prioritized by number of nodes from input list among all nodes in the network, *p*-value and z-score [95]. The algorithm output was a list of 30 networks, named by the linked TFs. We examined the top five significant TF networks (with the largest number of interconnections within the dataset but not necessarily included in it), and chose to investigate the most significant network named after the TF tethering it: ETS1.

## 4. Conclusions

This paper describes a proteomic investigation to identify the phosphorylated amino acid residues (PAARs) and associated proteins of the polyomavirus SV40 DNA replication initiation complex. While confirming many previously known PAARs, results from this study have provided a plethora of previously unidentified PAARs on the key cellular DNA replication/repair/DNA damage response complexes, PolPrim and RPA, as well as on the SV40 LT protein, which is also involved in these pathways. As validation of the new PAAR sites identified, we showed that phosphomimetic mutation of one of the novel PAAR sites on SV40 LT first identified in this study results in a dramatic decrease in only the DNA replication function of LT, both biochemically (in vitro) and in cultured cells [50]. The rest of the hundreds of novel PAAR sites found in this report thus provide new openings for investigation of regulation of important interactions and associations that SV40 has with its mammalian host, with implications for devising future targeted therapeutic strategies.

Additionally, while confirming several previously known proteins associated with LT, PolPrim-, and RPA-complexes, our study also provides hundreds of associated proteins that were previously not known to be so associated. Bioinformatics-based analyses of their interrelatedness are consistent with known biological functions. Subsequent in silico analyses identified a key overconnected transcription factor, ETS1, that was absent in our original IPMS dataset. We investigated LT for association with Ets1 and discovered the two proteins to be strongly associated, independent of the presence of nucleic acids, thus implicating the association to be important for viral transcription or cellular transformation. This finding validates our network analyses, showing that additional proteins identified in similar networks have the potential to interact with one or more of the complexes and perform heretofore unknown functions that are important for PyV biology. Overall, we identified many previously unidentified PAARs and associated proteins to fuel further investigations, as well as identified multiple networks containing additional proteins that, based on our Ets1 results, may also be associated with one or more of the complexes investigated in this study.

## Figures and Tables

**Figure 1 ijms-25-04540-f001:**
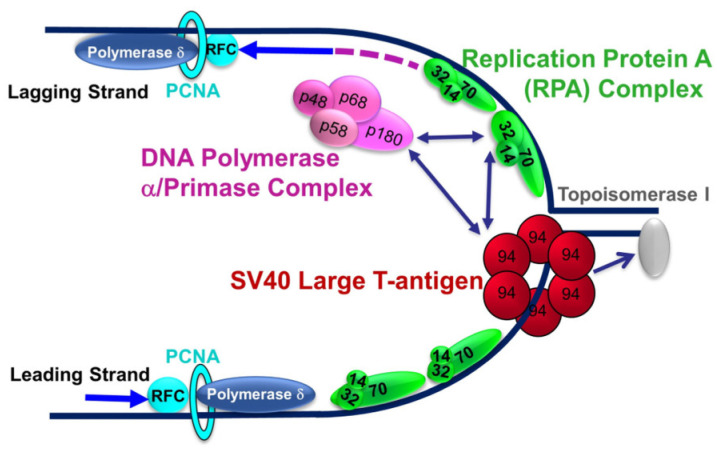
Schematic model of the elongating PyV DNA Replication Fork. Large T-antigen (LT) (monomer ~90 kDa; red) recognizes the origin of replication, and, after parking at the origin, acts like a helicase to begin unwinding the DNA (dark blue solid line) using ATP hydrolysis for energy. LT recruits the remainder of replication factors required for initiation and synthesis to drive replication of SV40 genomes from the host mammalian cell. The helicase along with Polprim (4 subunits, pink) and RPA (3 subunits, green) interact to begin the process of dsDNA unwinding at the fork and synthesis of RNA/DNA primers (dark pink dashed line on the lagging strand), the critical first steps in PyV DNA replication (bright blue arrows showing directionality). DNA Polymerase delta (blue), recruited through RFC/PCNA (teal), then replicates both the leading and lagging strands. DNA Topoisomerase 1 (TOP1, grey) acts ahead of the fork to relieve super helical tension that would hinder the unwinding of the parental DNA strands.

**Figure 2 ijms-25-04540-f002:**
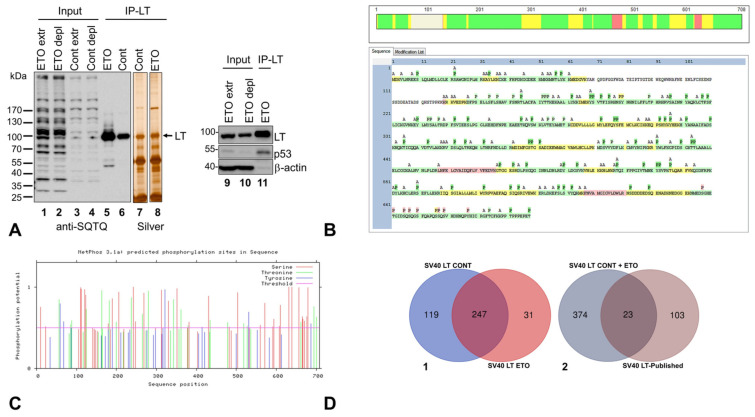
The coIP-ed proteins and PAARs on SV40 LT. IP experiments were performed with SV40 LT-specific monoclonal antibodies from whole cell extracts: untreated (Cont) and etoposide-treated (ETO) HEK293T cells. (**A**) Immunoblotting of whole- (extr) and IP depleted- (depl) cell extracts (Input) and eluted fractions (IP-LT) (lanes 1–6) were probed with anti-SQ/TQ antibodies. The IP-eluates were stained with silver (lanes 7, 8) and immunoblots of input and IP-eluate probed with specific antibodies to LT, p53 and β-actin (lanes 9–11). The protein p53 co-IPs with LT (lane11). (**B**) PAARs on SV40 LT amino acid sequence using MS/MS analysis. Only one representative sequence map with modifications for SV40 LT is shown since there was a large overlap between Cont and ETO. Sequence coverage of detected peptides (by MS) was 91% for both treatments. Tryptic peptide K67-K129 remained undetected by MS/MS (For further details of SV40 LT PAARs, see Appendix A). The tryptic peptide detection and S/T/Y phosphor-sites (P) are highlighted with colors by confidence levels (scale bar graphic on top of the full sequence map): green (high), yellow (medium), red (low), and white (undetected by MS/MS). To avoid overinterpretation of the data, the spontaneous O and A modifications of free amine groups are not included in this report (see Section 3). Abbreviations: P = phosphorylated residues, A = acetylated residues. Figure 2B is also included as Appendix A for further clarity. (**C**) Graphical output of in silico predictions of phosphorylated S, T, and Ys on SV40 LT protein sequence of 708 amino acids (aa). X-axis: number of aa residues; Y-axis: measure of the phosphorylation potential on a scale of 0 to >1 with a cut-off at ≥0.5 (magenta line) for positive predictions. (**D-1**) A Venn diagram of the SV40 LT co-immunoprecipitated proteins for Cont and ETO, to study the overlap between them (**D-2**). We compared the Gene IDs of nonredundant coIP-ed proteins with LT (IPMS) to the SV40 LT-associated proteins that were previously published to create a Venn diagram of overlap between the two groups.

**Figure 3 ijms-25-04540-f003:**
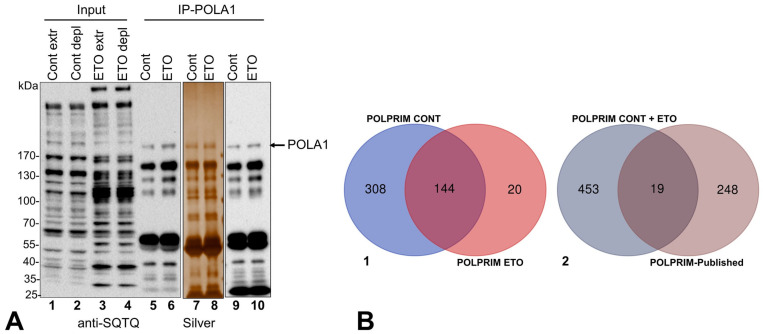
(**A**,**B**). The coIP-ed proteins with Polprim-complex. (**A**) The proteins from IP experiments with POLA1-specific monoclonal antibodies were resolved by gel electrophoresis as described. Immunoblots of whole- (extr) and IP depleted- (depl) cell lysate (Input) and IP fractions from Cont and ETO treatments (IP-POLA1) (lanes 1–6) were probed with anti-phospho-SQ/TQ antibodies. IP eluates from Cont and ETO were stained with silver (lanes 7–8) as well as immunoblotted (lanes 9–10) using anti-POLA1 antibodies. (**B-1**) coIP-ed proteins with Polprim-complex: A Venn diagram was created for the coIP-ed proteins identified by MS2 analysis to examine overlap between Cont. and ETO. (**B-2**) The Gene IDs of the nonredundant associated proteins with Polprim-complex identified by IPMS were compared to the Polprim-associated proteins published previously to create a Venn diagram of overlap between the two groups.

**Figure 4 ijms-25-04540-f004:**
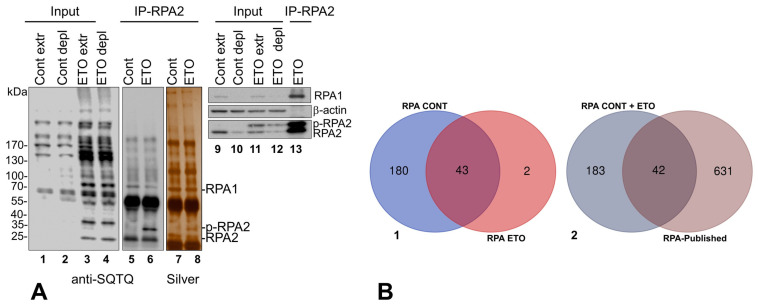
(**A**,**B**). The coIP-ed proteins with RPA-complex. (**A**) The proteins from IP experiments with RPA-specific monoclonal antibodies were resolved by gel electrophoresis. Immunoblot using anti-phospho-SQ/TQ antibodies of whole- (extr) and IP depleted- (depl) cell extracts (Input) and IP fractions (IP-RPA2) from Cont and ETO treatments (lanes 1–6). The IP elution lanes were stained with silver (lanes 7, 8). Immunoblots were probed by specific antibodies to RPA1, β-actin, and RPA2 (lanes 9–13). The p-RPA2 was clearly identified in the ETO treated samples over Control. The three bands are marked in lane 8. (1) RPA1 (70 kDa), (2) p-RPA2 (~35 kDa) and (3) unphosphorylated RPA2 (32 kDa) were corroborated by immunoblotting using specific antibodies (lanes 11–13). (**B-1**) A Venn diagram was created for the coIP-ed proteins with RPA identified by MS analysis to examine overlap between Cont. and ETO. (**B-2**) Gene IDs of coIP-ed proteins identified by IPMS were compared to the RPA-associated proteins previously published to create a Venn diagram of overlap between the two groups.

**Figure 5 ijms-25-04540-f005:**
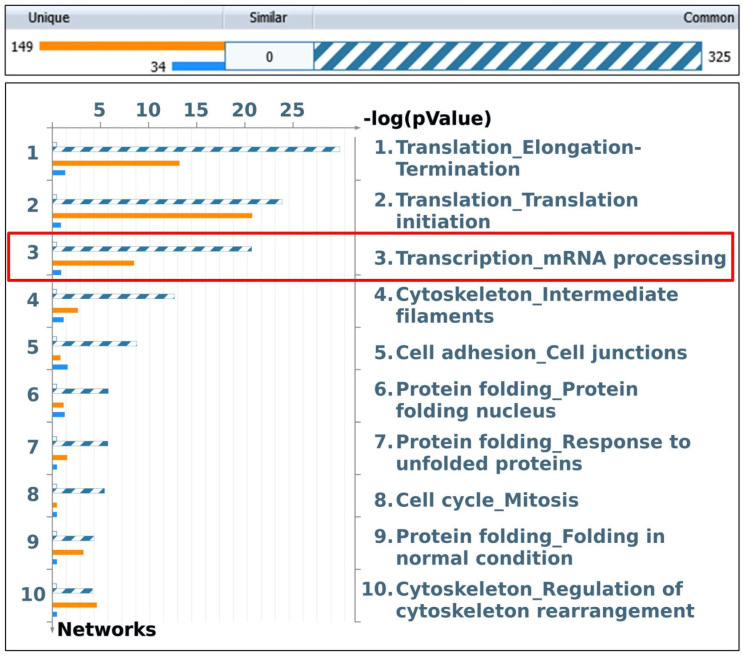
GO Enrichment analysis of SV40 LT-associated proteins. Gene content was aligned between all SV40-LT-associated proteins or network objects (NOs) identified by IPMS within the web-based integrative software suite (MetaCore). GO enrichment analysis shows the top 10 enriched process networks that are arranged in −log (pValue) from most to least relevant. This analysis shows transcription*_mRNA* processing (process network) to be the third most enriched among the top 10 (red box). The top two translation-related process networks are due to the presence of several coIP-ed ribosomal proteins (RP). RP aggregates are often found in IPs from lysates of robustly growing cells or cells in stress response [74].

**Figure 6 ijms-25-04540-f006:**
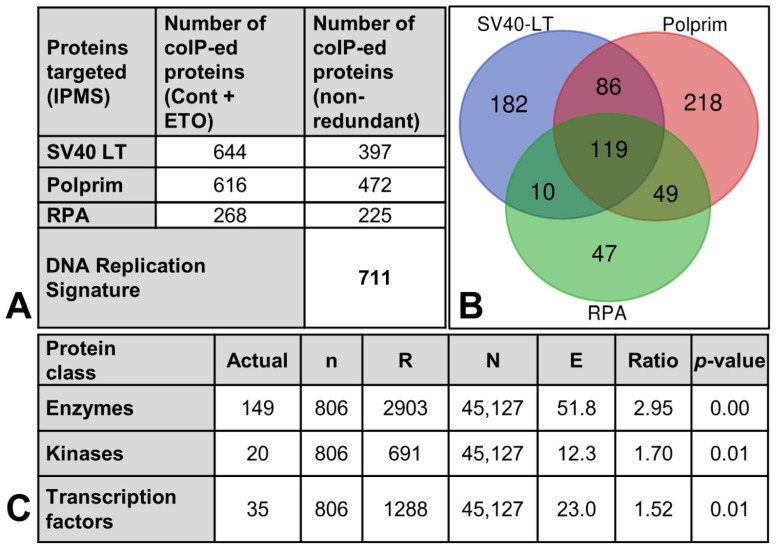
Global proteomic analyses. (**A**) Seven hundred and eleven nonredundant proteins (gene IDs) were found to coIP with either one/two/all three factors that are needed for initiation of PyV DNA replication: LT, Polprim-, and RPA-complex, and termed the “DNA replication signature”. (**B**) A Venn diagram shows the distribution and overlap of the 711 gene IDs between the three factors. (**C**) A GO enrichment analysis in MetaCore based on protein function scores ranks the three most relevant protein class related to this curated experimental dataset. Explanation of each column: Protein class: a broadly defined protein function; Actual; number of network objects from the activated dataset for a given protein class; n: total number of network objects in the activated dataset; R: total number of network objects of a given protein class in the complete database or background list; N: total number of network objects in the complete database or background list; E: # of objects that would be expected to occur by chance, mean value for hypergeometric distribution (n*R/N); Ratio: connectivity ratio (actual/expected); *p*-value: probability to have the given value of actual or higher (or lower for negative z-score).

**Figure 7 ijms-25-04540-f007:**
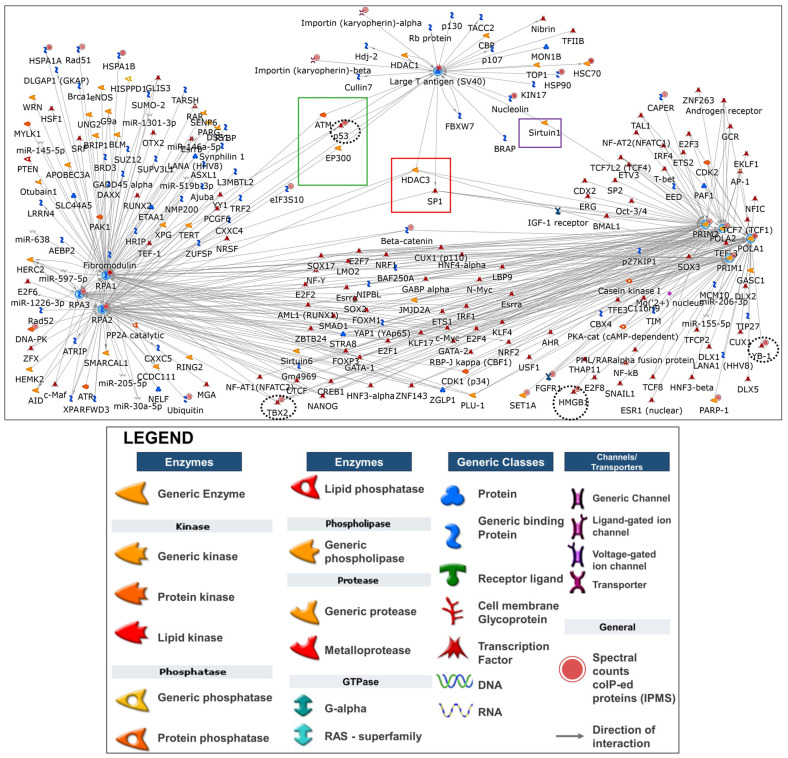
With legend: Interactome analysis. Interaction networks were unique for our dataset, built around the three protein factors (Gene IDs) with their respective subunits acting as “hubs”: (1) SV40 LT, (2) tetrameric Polprim-complex, and (3) trimeric RPA-complex (blue circles). In this network, all three factors are connected to each other via two proteins: (red box). LT is connected to RPA via three proteins: (green box) and to Polprim-complex via a single protein (purple box). Twenty-seven network objects (gene IDs) included in this network are from within our experimental dataset and are associated with solid red circles, of which four are TFs (black dashed circle). LT was associated with 30 overall interactions in this analysis, while RPA-complex (with RPA1: 92, RPA2: 75, RPA3: 29 interactions) and Polprim-complex (with POLA1: 71, POLA2: 54, PRIM1: 35, and PRIM2: 51 interactions) demonstrated higher interconnectivity to genes (from within or outside our experimental dataset). Information for all networks: Individual proteins or objects are represented as nodes, and different shapes of the nodes represent functional classes of proteins and are clarified in the legend. While the enrichments and networks building statistics are calculated according to our uploaded experimental dataset without spectral counts (SCs), the SCs are used to set thresholds to allow visualization of the semiquantitative measure for coIP-ed protein abundance in the IP eluates or cell lysate (solid red circles associated with network object). The network of interactions (also called edges: grey arrows) between proteins (with direction) were based on the curated knowledge base within Metacore. The arrowheads indicate the direction of the interactions. Large T-antigen was the only SV40 specific protein added to every network analysis. It mapped to a generic protein in Metacore, to which the red solid circle was added, the color consistent with spectral counts in our MS results.

**Figure 8 ijms-25-04540-f008:**
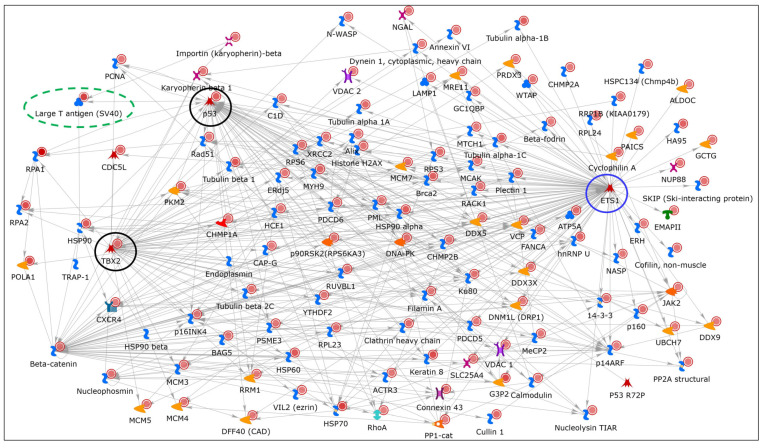
ETS1 is a key transcriptional factor. We selected the most highly significant “transcriptional regulation network” overconnected to our dataset, named for the TF (ETS1) tethering it, that was enriched with proteins from within our DNA replication signature (network objects with solid red circles). The merged network is redrawn after adding Large T-antigen manually (dashed green oval). ETS1 (blue circle) with 106 curated interactions is predicted to be the most significantly overconnected transcription factor, although it was unidentified within our IPMS dataset (“hidden”). For statistics, see Appendix A, sheets 1–11. Types of objects are clarified, and general information for all networks are in the legend and description of Figure 7.

**Figure 9 ijms-25-04540-f009:**
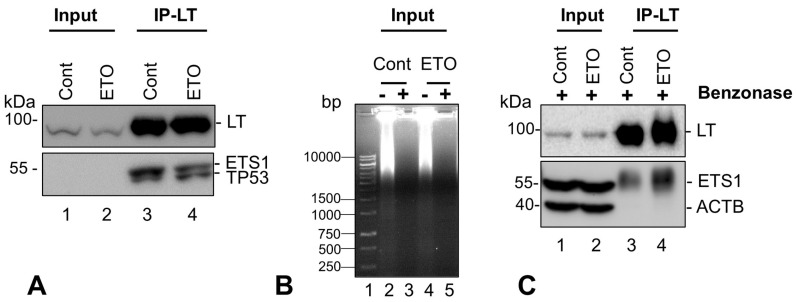
Validation of association of LT and Ets1(ETS1) +/− Benzonase treatment: (**A**) Immunoprecipitation experiments were performed with HEK293T whole cell lysates (Cont and ETO: Input). Immunoblots of Input and IP-LT lanes were probed using antibodies to Large T-antigen (LT) (upper panel), Ets1 (Novus), and p53 (lower panel) (lanes 1–4) as described in Methods. The mouse mAb to Ets1 (Novus) identified coIP-ed Ets1 in lanes 3, 4 but was unable to identify it in the Input lanes (lanes 1, 2). (**B**) To examine the effects of DNA on the association of LT with Ets1, nucleic acids were degraded in the whole cell lysates (Input) by Benzonase nuclease treatment before IP experiments were performed. Input lanes (without [−] and with [+] Benzonase) were run on an agarose gel stained with ethidium bromide. (**C**) Immunoblots of the Benzonase-treated cell extracts (Input) (lanes 1, 2) and corresponding IP-LT (lanes 3, 4) were probed with LT, Ets1 (Fisher), and β-Actin antibodies. The Ets1 rabbit mAb (Fisher) used here detects Ets1 in both input and IP lanes.

## Data Availability

The raw data supporting this report are included as Excel tables in the Appendix A. The sequencing data are uploaded and published in Mendeley data: DOI:10.17632/8srzx429ms.1; https://data.mendeley.com/datasets/8srzx429ms/1, accessed on 4 April 2024.

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
