# Peer review of "Proteomics Analysis of the Polyomavirus DNA Replication Initiation Complex Reveals Novel Functional Phosphorylated Residues and Associated Proteins"

_ijms, 2024, doi:10.3390/ijms25084540_

Round 1

Reviewer 1 Report

Comments and Suggestions for Authors

The manuscript entitled “Proteomics analysis of the polyomavirus DNA replication initi-2 ation complex reveals novel functional phosphorylated resi-3 dues and associated proteins” by Rama Dey-Rao and the colleagues. This study aimed to utilized the high-resolution liquid chromatography tandem mass spectrometry (LC-MS/MS) to explore the potential post translational phosphorylation sites on the Large T-antigen (LT), DNA polymerase alpha-pri-15 mase (Polprim), and single-stranded DNA binding complex (RPA) formed replication complex, and investigated immunoprecipitated factors (IPMS) associated with the replication complex. Results showed that 479 novel phosphorylated amino acid residues are identified on the three core components of the replication complex, and 374, 453, 183 novel proteins associated with the LT, Polprim, and RPA, respectively. Additionally, the author proved that ETS proto-oncogene 1 (ETS1), which is not identified by IPMS, plays significant roles in viral transcription and cellular transformation. All these results demonstrated that post translational phosphorylation modifications on the three core components (LT, Polprim, and RPA) of the replication complex play essentials roles in regulation of the virus replication, and underline same significant of the host factors involved. The MS supplied the clues for further investigation of virus replication from aspects of host factors or post translational phosphorylation. However, several issues need to be addressed prior to publication, as listed below:

1.     In Abstract, no any sentence to descript the significance of your research or the value to your research field. What is the innovation or advanced progressed of your MS.

2.     Why select the ETS proto-oncogene 1 (ETS1) for functional validation? OR why you not select a factor that identified from you IPMS by proteomics? If you chose the factor that from your IPMS by proteomics, which also demonstrate that your experiment is scientific and the results are creditable by proteomics. I think it is better to include two factors at least for functional analyses in viral replication for validating the creditable of your proteomics, such as TBS2 OR P53.

3.     Fig 2B, I think it is better to put it in supplementary files or we should enlarge it for reading and understanding.

4.     In part of data availability, the author should upload the sequencing data to a public database. 

Reviewer 2 Report

Comments and Suggestions for Authors

Overview and general recommendation:

During polyomavirus (PyV) infection, polyomavirus Large T-antigen (LT) conducts primer synthesis by interacting with DNA polymerase alpha-primase (Polprim) and single-stranded DNA binding complex (RPA). In the manuscript, the authors identified phosphorylated amino acid residues (PAARs) in LT, Polprim and RPA with immunoblotting and mass spectrometry. Then they performed GO enrichment analysis with all three sets of data. After removing the redundant genes, 711 genes are described as “DNA Replication Signature” and following interactome analysis, transcriptional regulation network analysis are performed with them.

Overall the manuscript is OK. I find the paper is well organized. The author performed detailed background research. It is well written and the figures are presented in an appropriate way. I think the authors should include more data describing how identified genes affect cell events and protein function.

Major comments:

1.      In this paper, only one example, ETS1 is included as validation of their analysis. I suggest the author should include at least 3 genes as validation and their effects on cellular function should be described.

Round 2

Reviewer 1 Report

Comments and Suggestions for Authors

No